# The Influence of Personality on Interpersonal Emotion Regulation in the Context of Psychosocial Stress

**DOI:** 10.3390/ijerph20043073

**Published:** 2023-02-09

**Authors:** Robin Wickett, Nils Muhlert, Karen Niven

**Affiliations:** 1Alliance Manchester Business School, University of Manchester, Manchester M15 6PB, UK; 2Division of Psychology, Communication & Human Neurosciences, University of Manchester, Manchester M15 6PB, UK; 3Sheffield University Management School, University of Sheffield, Sheffield S10 1FL, UK

**Keywords:** interpersonal emotion regulation, emotion regulation, personality, big five, trier social stress test

## Abstract

Interpersonal emotion regulation is common in everyday life and important to various outcomes. However, there is a lack of understanding about the personality profiles of people who are good at regulating others’ emotions. We conducted a dyadic study, pairing 89 ‘regulators’ and ‘targets’, with the targets subjected to a psychosocial stressor in the form of a job interview, and the regulators instructed to manage the targets’ feelings prior to the interview. We did not observe any relationship between the regulators’ personality traits and the strategies that they reported using when trying to manage the targets’ feelings, nor between the regulators’ personalities and the targets’ job interview performance. However, the anxiety levels of the targets who were paired with more extraverted regulators fluctuated less across the multiple measures throughout the study, suggesting more effective interpersonal emotion regulation. Our findings suggest that extraversion may be the most relevant trait in shaping interpersonal emotion regulation, and that the influence of personality on regulatory effectiveness is unlikely to arise due to preferences for using different types of strategies.

## 1. Introduction

People try to influence the feelings of others in many of the relationships they have, including with their friends, family members, workmates, and even virtual strangers. Over the past two decades, interest in this process of ‘interpersonal emotion regulation’ has grown, through a deeper appreciation of the social nature of emotion regulation [1]. Whereas *intra*personal emotion regulation concerns efforts to regulate one’s own emotions, *inter*personal emotion regulation describes deliberate, controlled attempts to maintain or change the direction or intensity of another person’s feelings [2]. Although a compelling body of empirical research has emerged linking successful interpersonal emotion regulation to desirable outcomes (e.g., higher quality relationships [3,4]; better personal well-being [5]; and enhanced performance in work and sports teams [6,7]); less is known about *who* is most successful at interpersonal emotion regulation. Therefore, the goal of this study was to assess whether particular aspects of the regulator’s personality influenced the process of interpersonal emotion regulation in the context of a psychosocial stressor.

The process of interpersonal emotion regulation involves four constituent tasks [8], each of which may be influenced by individual differences, such as regulator personality [9]. First, the regulator must accurately identify the underlying emotional state of the target. Second, the regulator must set a regulatory goal and evaluate whether regulation is needed to achieve this goal. Third, the regulator must generate and select the most appropriate strategy to achieve their regulatory goal. Last, the regulator must implement their chosen strategy, a process which, if effective, results in the intended change in the target’s emotions and potentially in fulfilling higher-order goals (e.g., enhancing the target’s performance). In this study, we examined how the latter two stages (i.e., those most proximate to the outcome of interpersonal emotion regulation) relate to a regulator’s personality.

The present study, therefore, had two aims. The first aim was to investigate whether and how a regulator’s personality influences their choice of interpersonal emotion regulation strategies. That is, in a situation where a regulatory goal is specified, does the regulator’s personality affect which strategy they select to try and achieve that goal. The second aim was to investigate whether and how the personality of a regulator influences their effectiveness during the implementation of regulation. We adopt the trait approach to personality to address the aims of our research. The trait approach is a useful perspective for exploring the influence of personality in a new domain, such as interpersonal emotion regulation, because it describes personality in terms of broad units of analysis (i.e., traits). We centre our research on the Big Five framework [10], the most well-established classification of personality traits, which describes individuals’ stable dispositions to think, feel, and behave along five continua: extraversion, agreeableness, conscientiousness, neuroticism, and openness to experience. Given the strong evidence that the traits of extraversion, agreeableness, and neuroticism are the most salient in explaining individual differences in intrapersonal emotion regulation (e.g., see the review of Hughes et al. [11]), we focus on these traits in our study.

Our study used an adapted version of a well-established psychosocial stress paradigm, the Trier Social Stress Test (TSST) [12]. In the TSST, the participants are required to perform a job interview, followed by a surprise mathematical test, in front of an unresponsive interview panel. Our key adaptation was to run the paradigm with a period of interpersonal emotion regulation included immediately prior to the social stressor, so that we could explore the influence of the regulator’s personality on their interpersonal emotion regulation strategy selection and the effectiveness of their regulation.

### 1.1. Selection of Interpersonal Emotion Regulation Strategies

The research to date examining personality and interpersonal emotion regulation has focused on the associations with strategy use, reporting links between personality and the tendency to use strategies to either improve or worsen others’ feelings [13,14]. However, there are distinctive strategy types that people can use when seeking to improve (or worsen) the feelings of others, which the research suggests are differentially effective [6,15], meaning that a more nuanced understanding of the relationship between personality and the selection of specific strategies is important.

There are two main theoretical frameworks that have been proposed for differentiating interpersonal emotion regulation strategies. First, the application of Gross’s process model [16] of emotion regulation to the interpersonal domain distinguishes the strategies based on whether they deal with the underlying causes of a target’s emotions (*problem-focused*), or whether they concentrate on managing the target’s emotional reaction (*emotion-focused*, e.g., [17,18]). The problem-focused strategies include the selection and modification of situations that cause emotions and the cognitive reappraisal of situations that cause emotion. The emotion-focused strategies include attentional deployment towards or away from a stimulus to manage the emotional response and response modulation, which describes the suppression or exaggeration of emotional responses.

Second, Niven et al. [19] propose a key distinction between engagement and relational strategies. Engagement strategies, later labelled as *cognitive* [15,20], involve changing another person’s thoughts in order to change their feelings (and so are akin to cognitive reappraisal from Gross’s framework). In contrast, relational or *socio-affective* strategies involve communicating a message about one’s relationship with the target (e.g., a message of comfort, care, and validation) in order to change their feelings. The research indicates that socio-affective strategies (which are not included in Gross’s [16] framework) have distinctive effects to other strategy types, for example, being stronger predictors of positive relational outcomes, such as feelings of closeness [20], and the formation of new relationships [21]. Thus, in the present research, we explore how personality affects the use of strategies from Gross’s model and socio-affective strategies. Note that, because the situation in which targets are placed is fixed in our study, we do not include situation selection or modification.

### 1.2. Personality and Selection of Strategies

People who are high in agreeableness are characterised by their desire to maintain positive relationships with others [22]. They are also more likely to experience distress when faced with potential conflicts, as they are highly sensitive to the feelings of others [23]. Highly agreeable individuals should therefore prefer using socio-affective strategies because they ought to maximise social harmony and minimise conflict during interpersonal emotion regulation [20,24]. In contrast, strategies that are cognitive in nature may be less attractive to those high in agreeableness, due to the possibility of evoking conflict. From the target’s perspective, rather than the regulator validating and reaffirming their world view, a cognitive strategy challenges it (*Why don’t you think about situation x in a different way?* [21,25]). Similarly, strategies that involve response modulation (e.g., telling a target to ‘cheer up’ or ‘calm down’) may come across as being abrupt or confrontational (e.g., they are associated with increased blood pressure in targets [26]), and may therefore also be avoided by those with high agreeableness.

Like the people high in agreeableness, individuals who are high in extraversion value social experiences [27] and cultivating satisfying relationships [28,29]. Highly extraverted individuals are therefore likely to favour socio-affective strategies because these should promote their social goals [20,21]. However, in contrast to individuals who are high in agreeableness, who have concerns about coming across as confrontational or stimulating conflict in their interactions with others, individuals who are high in extraversion display a greater degree of dominance in their relationships and may not shy away from more confrontational strategies [24]. This social dominance, combined with high self-efficacy in their ability to regulate emotion [30], suggests that those high in extraversion will be more likely to engage in proactive strategies that attempt to address the underlying causes of a target’s emotions [31], just as they do when managing their own feelings [11]. Thus, regulators who are high in extraversion may also be more likely to use cognitive reappraisal.

Neuroticism involves negative emotionality, behavioural inhibition, and disproportionate responses to stress [32]. Because individuals who are high in neuroticism react more intensely to aversive or negative stimuli, they typically seek to immediately reduce exposure by using avoidance and disengagement strategies when regulating their own feelings [33,34]. The negative emotions of others, such as anxiety, can be considered a negative stimulus that high-neuroticism individuals would ordinarily seek to limit exposure to. When dealing with the negative emotions of others in a situation they cannot easily avoid or disengage from, individuals who are high in neuroticism may therefore choose to down-regulate the targets’ feelings by engaging in emotion-focused strategies (i.e., attention deployment and response modulation). Such strategies deal with the ‘problem’ of their own exposure to a negative stimulus (i.e., the target’s negative emotion) in the most direct manner and therefore ought to be favoured.

**Hypothesis 1 (H1).** 
*The agreeableness of the regulator will be (a) positively associated with the use of socio-affective strategies, (b) negatively associated with cognitive reappraisal strategies, and (c) negatively associated with response modulation strategies.*


**Hypothesis 2 (H2).** 
*The extraversion of the regulator will be positively associated with the use of (a) socio-affective and (b) cognitive reappraisal strategies.*


**Hypothesis 3 (H3).** 
*The neuroticism of the regulator will be positively associated with the use of (a) attention deployment and (b) response modulation strategies.*


### 1.3. Implementation of Interpersonal Emotion Regulation

During implementation, effectiveness can be determined based on whether the regulator’s desired outcomes are induced. This combines inducing the desired effects in relation to the target’s emotions (the proximate goal of interpersonal emotion regulation) and inducing the associated desired consequences in relation to the higher-order goals of regulation [35,36]. The current paradigm involved an explicit instruction to regulate a target’s feelings to help them to perform optimally in an interview; thus, implementation effectiveness takes the dual form of managing the target’s anxiety (which would ordinarily be expected to be experienced prior to and during an interview, and which could undermine performance, e.g., [37]), and enhancing the target’s interview performance.

### 1.4. Personality and Implementation Effectiveness

People who are agreeable tend to pay greater attention to the social cues of others [38]. This may mean that they are relatively adept at identifying the optimal time to implement their chosen regulation approach and responding to social feedback from the target when judging if a regulation attempt has been successful or needs rethinking [8,9]. However, the strategy type we theorised to be favoured by those high in agreeableness—socio-affective—is not typically found to be especially effective when it comes to eliciting the desired outcomes of interpersonal emotion regulation, such as a change in affect or performance [6,17], that we focus on here. Thus, it is not clear whether regulators who are high in agreeableness will be particularly effective at implementing interpersonal emotion regulation.

Regulators who are high in extraversion will have a natural tendency to experience positive affect, particularly drawing from their positive energy when interacting with others [39]. They may therefore be more skilled at implementing regulatory efforts that entail improving the feelings of others, as their affective tendencies align with the goals of their regulation attempts, making the act of regulation easier to achieve. For example, the positive affect and energy that they experience during social interaction may be ‘caught’ by the regulatory target through a process of emotional contagion [40]. Our expectation that those high in extraversion would be likely to use the problem-focused strategy of cognitive reappraisal further suggests that extraverts may be more effective in terms of implementing interpersonal emotion regulation. This is because cognitive reappraisal has been reported to be a particularly effective strategy when it comes to reducing negative affect [6,20] and enhancing performance in others [17]. As such, we would expect regulators high in extraversion to be more effective at inducing the desired outcomes in others.

Finally, we expect that those high in neuroticism may be less effective in their implementation of regulation due to their self-focus. For example, because individuals high in neuroticism typically tend to avoid negative emotions in themselves [41], they may implement interpersonal emotion regulation before it is optimal to do so [8] to avoid exposure to negative affect in others. Those high in neuroticism may also be less attuned to the social cues presented by regulatory targets [42], and therefore less able to determine if a regulation attempt is going poorly, and a change of approach is needed. Further supporting the perspective that those high in neuroticism may be poor at inducing the desired outcomes in others, the strategies we expect to be favoured by those high in neuroticism (i.e., attention deployment and response modulation) have been shown to be ineffective in terms of changing others’ affect and enhancing others’ performance [6,17].

**Hypothesis 4 (H4).** 
*The extraversion of the regulator will be positively associated with the amount of anxiety experienced and performance of the target during the interview.*


**Hypothesis 5 (H5).** 
*The neuroticism of the regulator will be negatively associated with the target’s levels of anxiety and performance during interpersonal emotion regulation.*


## 2. Materials and Methods

### 2.1. Design

We designed an adaptation of the well-established Trier Social Stress Test (TSST) [12]. In this task, psychosocial stress is induced by asking participants to undertake an interview for a job (in this case, working in the School of Psychology), followed by an unexpected challenging mathematics task that involves forced failure. Both tasks are carried out in front of two peers providing neutral feedback (and so eliciting psychosocial stress). In our adaptation, the participants were placed into pairs, with one participant assigned to the role of ‘regulator’ and one to the role of ‘target’. In each study session, the pair attended at the same time, and the regulator participant was instructed to manage the feelings of the target participant, who underwent the TSST. This provided a context that involved a clear opportunity and motivation for the use of interpersonal emotion regulation.

The study was conducted in an online virtual environment using the online platform GatherTown (https://www.gather.town, accessed on 27 March 2022), which is a virtual video-calling space that allows multiple parallel conversations to flow freely. All the participants logged in using their own personal credentials and were represented by a unique virtual avatar, which they could manoeuvre around rooms within a virtual university campus. The study was conducted in accordance with the Declaration of Helsinki and approved by the institutional ethics committee.

### 2.2. Participants

One-hundred and seventy-six undergraduate participants on a psychology degree programme were recruited from a UK university in exchange for course credits. The participants were recruited to partake in one of two studies, which were described in identical terms as being about interviews and memory. In reality, the studies were part of the same piece of research, and the two studies were used in order to assign people to the role of either regulator or target. The participants who signed up for one study were excluded from the other to ensure we did not have individuals in the sample who had participated twice during different study sessions. The inclusion criteria required all the participants to have no history of neurological or psychiatric disorders. The sample was heavily represented by females (76.4%) compared to males (23.6%), with all the participants being within the 18–23 age bracket (*M*_age_ = 19.58 years, *SD* = 1.51). These demographics are typical of UK undergraduate psychology courses [43].

### 2.3. Procedure

The study involved a session in which both the regulator and target participants took part in parallel. The order of the tasks is illustrated in Figure 1.

#### 2.3.1. Pre-Regulation Period

When the ‘regulator’ participants arrived at the study session, they were given a short questionnaire battery, including a self-reported measure of their personality. They were then led to a waiting room by one of the research team.

When the ‘target’ participants arrived at the study session, they completed a first measure of their state anxiety, then were led to meet the interview panel, which consisted of two trained confederates, both of whom were students in a higher level of study than the participants. During this meeting, the targets were told by the interviewers that they would have to deliver a speech for a job position that they were applying to at the School of Psychology. After the target had met the panel, the member of the research team took them to the waiting room to meet the regulator.

#### 2.3.2. Regulation Period

When the regulators entered the waiting room, they were informed that they would soon be joined by another person (the target), who would be preparing for a job interview, and given instructions to regulate that person’s feelings. All the regulators were given the same instruction: “Your job is to try and manage the feelings of the target so that they perform optimally in the interview.” The regulation period (beginning when the target entered the waiting room) lasted for 5 min.

#### 2.3.3. Post-Regulation Period

Immediately following the regulation period, the regulator completed a post-experiment questionnaire, responding to measures relating to the strategies they had used to influence the target’s feelings. After completing this questionnaire, the regulators were debriefed. Meanwhile, the target was left for a further 5 min to prepare for the interview. Following the 5 min preparation period, the target completed a second state anxiety rating.

#### 2.3.4. Social Stressor

The targets were taken to the interview room, where the interview panel was waiting for them. They were then asked to explain “Why would you make a good candidate for the job?” If the target stopped speaking before the 5 min expired, the interviewers would wait 20 s before following up with subsequent questions: “What kinds of jobs have you had in the past?”, and “Can you think of a situation where you have had to come up with new ideas to form a solution to a problem?” The interviewers were trained to remain in a neutral expression throughout. Both members of the interview panel completed evaluation sheets wherein they rated the target’s interview performance. After the interview, the targets completed a third measure of their state anxiety, and then undertook the mathematics task (“count backwards in 13 s from 1667”) in front of the interview panel, who remained in a neutral expression, followed by a fourth state anxiety measure. They were then asked to wait for a further 5 min before being given a final state anxiety measure and then being debriefed.

### 2.4. Measures

#### 2.4.1. Regulator Measures

##### Personality

The students who were assigned as the regulators were asked to complete John and Srivastava’s [44] Big Five Inventory at the start of the study session. This measure contains items for each of the Big Five domains of personality of interest: extraversion (eight items, e.g., “is talkative”, α = 0.87), agreeableness (nine items, e.g., “is helpful and unselfish with others”, α = 0.71), and neuroticism (eight items, e.g., “worries a lot”, α = 0.85). For each item, the participants were asked how much each one accurately described them. The scale was answered on a 7-point scale, where 1 = disagree strongly and 7 = agree strongly.

##### Interpersonal Emotion Regulation Strategy Use

The regulators indicated the extent to which they had used four types of strategies to regulate the targets’ feelings immediately after the period of regulation. For each strategy type, we reduced the number of items from the original scales to three in order to avoid over-exerting the regulators, in each case selecting the strategies with the highest item factor loadings from the original scale development. We used the scale stem: “During your interaction with the target, to what extent did you use the following strategies?”. We used the interpersonal emotion management scale [6] to capture the attentional deployment (e.g., “I distracted them from focusing on negative aspects of the interview.”; α = 0.66), cognitive change (e.g., “I tried to influence the emotions of the other person by changing how they thought about the interview.”; α = 0.71), and response modulation (e.g., “I suggested strategies for them to suppress undesirable emotions.”; α = 0.57). The low internal consistency reliability of the response modulation measure was moderately improved (to α = 0.66) with the removal of one item. However, the pattern and significance of the correlation and regression analyses with the two-item variable were unchanged. We report the analyses using the three-item variable. We also used Pauw et al.’s [15] measure of socio-affective strategies (e.g., “I comforted the other person.”; α = 0.87). All the items were answered on a 7-point Likert-type scale (where 1 = strongly disagree, and 7 = strongly agree).

#### 2.4.2. Target Measures

##### State-Level Anxiety

The state level of anxiety in the target was repeatedly measured throughout the study session using a 1–10 visual analogue scale, where the targets indicated how anxious they felt at that moment, from 1 (not very anxious) to 10 (extremely anxious). The timings of the anxiety measure (illustrated in Figure 1) were: directly before they met the interview panel, after the interaction with the regulator, after the interview, after the mathematics task, and after the five-minute wait towards the end of the study.

The target’s five self-reported ratings of anxiety were then compiled and converted into an index called the ‘area-under-curve’ (AUC). The AUC reflects changes in self-reported anxiety across time points by calculating the total area under the curve of all the measurements from each other (i.e., the change over time) and the distance of these measures from zero (i.e., the level at which the changes over time occur). AUC is calculated using the following equation, wherein *n* denotes the total number of anxiety measurements (in this case, 5 [45]) and *m_i_* denotes the single measurements:$$AUC_{1}=\left(\overset{n-1}{\underset{i=1}{\sum }}\frac{(m_{\left(i+1\right)}+m_{i}}{2}\right)-\left(n-1\right)\cdot m_{1}$$

AUC is considered a robust means of capturing the extent of fluctuation in anxiety during the TSST, which allows researchers to simplify statistical analyses and increase the power of tests without losing any information from the multiple measurements of anxiety [46]. A higher AUC score represents greater levels of fluctuation in anxiety across the rating points, whereas a lower score reflects a more stable trajectory between the target’s anxiety ratings. In the case of our study, a lower AUC would indicate more effective interpersonal emotion regulation, because the regulators were trying to reduce the anticipated increase in anxiety due to the psychosocial stressor.

#### 2.4.3. Interviewer Measures

##### Interviewee Performance

Immediately after the target had completed their interview, the two panel members were asked to discuss and agree on a rating for the target’s performance in the interview on a scale 1–10 (where 1 = very poor and 10 = very good). If the panel could not agree on a rating, they were asked to meet in the middle of their two scores.

### 2.5. Analysis Strategy

All the analyses were conducted in SPSS (version 27). As our interest was in the influence of personality on interpersonal emotion regulation, rather than in the influence of personality over and above other factors, we did not control for background variables, such as regulator and target gender, when testing our hypotheses [47,48]. We conducted a series of linear regressions, each with a unique dependent variable. We entered agreeableness, extraversion, and neuroticism as predictors simultaneously within each model in order to provide a more conservative test of our hypotheses. A post hoc power analysis using Soper’s [49] calculator revealed that, for a sample of our size (*N* = 89 regulators), with a three-predictor linear regression, our analytic strategy would have a power of 0.88 to detect a medium effect size of 0.15.

## 3. Results

Table 1 presents the means, standard deviations, and intercorrelations of all the study variables, including the background characteristics. The correlations show that regulator and target gender were largely unrelated to the outcomes of interest in our study, with the single exception of a negative association between regulator gender and cognitive reappraisal, indicating that female participants were less likely to adopt this strategy. The strategies used by the regulators were largely unrelated to interpersonal emotion regulation implementation outcomes, with the exception that response modulation was positively related to the targets’ anxiety AUC. In other words, when the regulators used more response modulation (e.g., telling the target to ‘cheer up’ or ‘stop worrying’), the targets experienced higher fluctuations in their anxiety, an indicator of regulatory failure.

We ran a series of four regressions, each with a different interpersonal emotion regulation strategy as the dependent variable, to test Hypotheses 1–3 (see Table 2). Contrary to the hypotheses, the regulators’ personality traits did not predict their use of any of the strategies.

To test Hypotheses 4–5, which proposed that the regulators’ personality traits would be associated with the effectiveness of their interpersonal emotion regulation implementations, we ran two further regressions, one testing the effects of the regulators’ personality on the targets’ anxiety AUC, and the other testing the effects on the targets’ performance during the job interview, as rated by the interview panellists (see Table 3). While none of the traits predicted the interview performance, contrary to Hypothesis 5, regulator extraversion was negatively associated with the targets’ anxiety AUC (b = −0.251, *p* < 0.05), in partial support of Hypothesis 4. The latter result indicates that the targets who were paired with a more extraverted regulator experienced less fluctuation in their anxiety, suggesting that their anxiety was better regulated.

Additional exploratory analyses attempted to pinpoint the specific times at which regulators’ extraversion was making a difference to the targets’ anxiety. We conducted a series of regressions with the raw anxiety measures (from the different time points across the study period) as the dependent variables, the baseline anxiety included as a control variable in Step 1, and the three personality traits as predictors in Step 2. The results, in Table 4, show that the targets who were paired with regulators who were higher in extraversion experienced lower levels of anxiety (controlling for the baseline) after both stressors (i.e., the interview and mathematics task). These results confirm that the anxiety of targets who were paired with more extraverted regulators was better managed during the psychosocial stressor they encountered, and that those targets experienced further protection against the subsequent mathematical stressor.

## 4. Discussion

We know surprisingly little about which types of people make the best regulators of others’ feelings. In the present study, we constructed a scenario where the participants were given an opportunity and goal to regulate the feelings of a peer who was about to undergo a psychosocial stressor, in order to explore whether and how the regulator’s personality would affect their interpersonal emotion regulation.

A key finding, contrary to expectations, was that the regulator’s personality was not related to the strategies that the regulator chose to use in this context. A possible reason for this is that the specific context that we constructed for regulation formed a particularly strong situation. According to interactionist views of personality, behaviour is the product of interactions between a person and context factors [49]. When a situation is strong, there are clear expectations or norms about how one should behave, and personality traits therefore become less important in explaining behaviour. Our paradigm may have modelled a relatively strong situation, resulting in a somewhat uniform approach to regulation across the entire group of regulators (as supported by the low standard deviations of the strategies relative to the 1–7 response scale; all the *SD*s were below 1). The null findings in this regard are important because the situation we adopted mirrored a realistic context in which interpersonal emotion regulation would be used (i.e., managing someone’s anxiety about an upcoming stressful event), meaning that our findings give us insight into the role of personality in interpersonal emotion regulation in a relatively common scenario. Specifically, they highlight that, in this type of situation, there is little personality-based variation in the strategies selected by regulators. Instead, regulators typically prefer to use socio-affective strategies and to avoid using response modulation (according to the mean scores).

Another possibility for why we failed to observe the expected relationships between regulator personality and the regulatory strategies they selected is that, in our study, the regulators and targets were strangers to each other. The research has indicated that people vary the strategies they use to regulate others’ feelings across the relationships they have [3], and that relational closeness is an important factor in shaping interpersonal emotion regulation strategies [50]. It could be that there is less diversity between individuals in the strategies they select to regulate the feelings of others when there is a lack of personalised information (stemming from relational closeness) about who those others are. However, we contend that our findings remain relevant, because people do try to influence the feelings of strangers or acquaintances [51,52], meaning that the scenario we study mirrors ‘real world’ interpersonal emotion regulation. Interpersonal emotion regulation towards strangers or acquaintances is particularly common in work situations; for example, those who work in service roles are expected to manage the feelings of their customers or clients [53]. The regulation of strangers’ feelings also occurs when people enter new social situations, such as new courses of study, new jobs, new sports teams, and so on [4,21].

A final possibility is that the pattern of null results in this regard could be an issue of range restriction stemming from the low observed variability in the traits of agreeableness and neuroticism in our sample. However, the means and standard deviations for personality traits in our sample are comparable to other studies sampling British undergraduate psychology students (e.g., [54]), meaning that our sample is likely to be representative of the wider population from which it is drawn.

Our findings therefore suggest that, at least in situations where people try to manage the anxiety of strangers or acquaintances who are facing stressful events, and at least within the population studied, personality plays little role in shaping the strategies people select. It is possible that, instead of regulator personality directly affecting strategy choice, as we had anticipated, such that particular traits align with particular strategies, personality rather shapes *how* strategies are selected. Perhaps, for example, because agreeable regulators are more attuned to the needs and responses of others, they adapt the strategies they use based on their knowledge of the person and the cues they gather during their interactions. Meanwhile, neurotic regulators may select whichever strategy they believe will be most effective in a given moment for shutting down the negative states of others, which might vary depending on the person they are dealing with and the situation they are in.

Although the regulators’ personality did not shape the strategies that the regulators selected, there was evidence that it influenced how good the regulators were at achieving their goal. We found that when the regulator was higher in extraversion, their partner experienced less fluctuation in their anxiety throughout the study—and specifically lower anxiety following the stressors they were exposed to—suggesting that extraverted regulators are more effective at reducing anxiety in their targets. Since we found no link between regulator personality and strategy use, it seems highly likely that this effect is driven by other differences relating to personality, such as the higher tendency of extraverted people to experience and express positive affect during interactions [55]. The finding that people who are higher in extraversion appear to be more able to deal with others’ anxiety suggests that such individuals may be well positioned in roles that involve this task, such as crisis intervention, debt management, counselling, and career coaching. However, it remains to be seen if extraverted regulators would be similarly effective if their goal was to increase negative affect in a target, or whether the enhanced regulatory implementation we observe here is restricted to situations where the goal is to make someone else feel better.

### Strengths, Limitations and Future Directions

The key strength of the present research is that we constructed a situation in which participants interacted with a real partner and we could observe the effects of their actions on their partner. We also did not manipulate or restrict the regulation behaviours that regulators could select and they were given a period of time in which they could use multiple strategies, which reflects how interpersonal emotion regulation occurs in daily life [53].

However, our design did include some important limitations. First, the situation we created was somewhat artificial, and it is yet to be determined how authentic and natural the participants’ responses were. Moreover, although we sought to simulate a realistic interview experience to the targets, there was not a great deal at stake compared to a real interview, which may have reduced the regulators’ motivation to engage with the study. Observing real interactions between people who know each other, such as friendship pairs, might reduce the strength of the situation and would enhance the ecological validity of our findings, allowing a clearer answer to whether personality does have a role to play in shaping people’s regulation strategy choices.

A second limitation is that we relied on the participants’ self-reports about which strategies they had used to regulate the targets’ feelings. Although we asked them to report these strategies immediately after the five-minute regulation period, limiting the extent to which retrospective recall biases ought to influence their reports, there is an open question of whether people are always consciously aware of the behaviours they use when regulating others’ feelings. An alternative approach would be to video-record the regulation interactions, so that the regulators’ behaviours could be independently coded. However, scholars have argued that interpersonal emotion regulation is defined based on the intention of the regulator (e.g., [2]; simply because a behaviour is observed does not mean that the regulator intended this as an act of regulation). Thus, future studies may wish to combine video-coding with self-reports, as used here, in order to triangulate the evidence. A further advantage of video-recording interactions is that we would gain insight into additional personality-related differences in interpersonal emotion regulation, aside from strategy use, that could explain why regulators high in extraversion are apparently more effective at regulating others’ anxiety (e.g., do they interact more with the target, or express more positive affect?).

A third limitation is that our sample size was relatively small for a study of individual differences. While our post hoc power analysis indicated that we had a sufficient sample size to detect medium-sized effects, the influence of personality on interpersonal emotion regulation may be more modest, and future studies involving larger samples may be helpful. A further opportunity for future research is to extend the way in which regulator personality is conceptualised, moving from the trait approach as used here, which captures broad features of personality, to a facet approach, focusing on narrow, unidimensional descriptions of personality. The research suggests that personality facets offer incremental power in predicting multiple criteria, so future studies might observe stronger links between personality and interpersonal emotion regulation by taking a more nuanced approach to personality. Further individual differences, such as those relating to the target of regulation (e.g., their general anxiety levels), could also be explored. A final suggestion for future research is to explore the possibility of non-linear relationships between personality and interpersonal emotion regulation.

## 5. Conclusions

In this study, we explored whether and how the personality of a regulator would influence their attempts at regulating the emotions of a target. We did not observe any relationship between the regulators’ personality traits and the strategies that they reported using to manage the targets’ feelings, nor between the regulators’ personalities and the targets’ job interview performance. However, the anxiety levels of the targets who were paired with more extraverted regulators fluctuated less throughout the study, suggesting more effective interpersonal emotion regulation. Future research should consider measuring personality at the facet level and coding behaviours during regulator-target interactions to better understand the relationship between personality and interpersonal emotion regulation.

## Figures and Tables

**Figure 1 ijerph-20-03073-f001:**
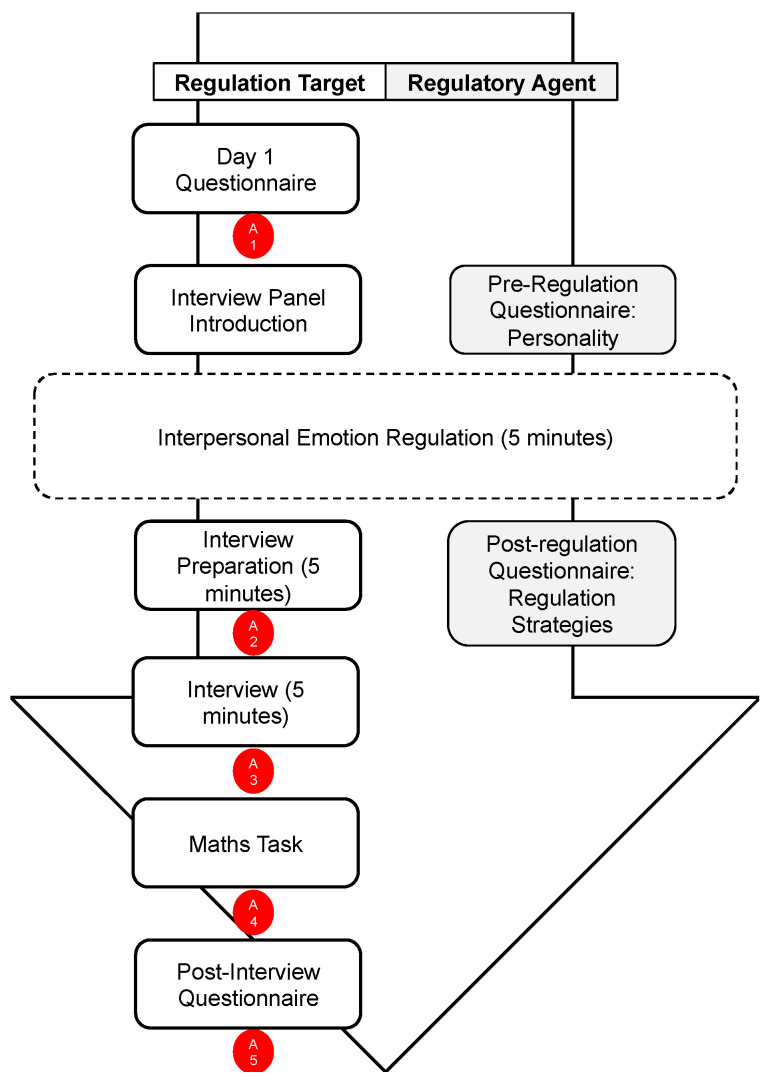
Timeline of study for both the regulator and target. Note: A1 = Anxiety Time 1; A2 = Anxiety Time 2; A3 = Anxiety Time 3; A4 = Anxiety Time 4; A5 = Anxiety Time 5.

**Table 1 ijerph-20-03073-t001:** Means, standard deviations, and intercorrelations between main study variables (*N* = 89 regulators, 89 target).

Variable	*M*	*SD*	1	2	3	4	5	6	7	8	9	10
1. Target gender (M, F)	Count: 16, 72	–	–									
2. Regulator gender (M, F)	Count: 21, 67	–	−0.082	–								
3. Regulator extraversion	4.55	1.07	0.020	0.119	–							
4. Regulator agreeableness	5.24	0.88	0.110	0.285 **	0.123	–						
5. Regulator neuroticism	4.31	0.95	−0.061	0.083	−0.382 **	−0.138	–					
6. Attention deployment	2.71	0.89	0.064	−0.075	0.141	0.110	−0.026	–				
7. Cognitive reappraisal	2.56	0.95	0.011	−0.286 *	0.163	0.030	0.053	0.612 **	–			
8. Response modulation	1.64	0.74	0.111	−0.205	−0.047	−0.102	0.029	0.332 **	0.346 **	–		
9. Socio-affective	3.15	0.95	0.195	0.059	0.196	0.139	0.018	0.565 **	0.587 **	0.290 **	–	
10. Target AUC anxiety	215.84	99.46	0.015	−0.182	−0.266 *	−0.070	0.128	0.025	0.113	0.234 *	0.012	–
11. Target interview performance	6.58	1.87	–0.033	−0.048	−0.112	−0.006	−0.003	−0.020	−0.037	−0.041	−0.082	−0.201

Note: * *p* < 0.05, ** *p* < 0.01. Both gender variables are coded, 1 = male, 2 = female.

**Table 2 ijerph-20-03073-t002:** Effects of regulator personality on strategy selection (*N* = 89).

	Attention Deployment	Cognitive Reappraisal	Response Modulation	Socio-Affective
	β	β	β	β
Agreeableness	0.145	0.022	−0.097	0.128
Extraversion	0.098	0.212	−0.034	0.227
Neuroticism	0.043	0.137	0.002	0.122
R^2^	0.030	0.042	0.012	0.064

* *p* < 0.05. ** *p* < 0.01.

**Table 3 ijerph-20-03073-t003:** Effects of regulator personality on implementation effectiveness (*N* = 89 regulators, 89 targets).

	Target Anxiety Area-Under-Curve (AUC)	Target Interview Performance Rating
	β	β
Agreeableness	0.027	0.003
Extraversion	−0.251 *	−0.133
Neuroticism	0.027	−0.053
R^2^	0.073	0.015

* *p* < 0.05. ** *p* < 0.01.

**Table 4 ijerph-20-03073-t004:** Effects of regulator personality on raw anxiety scores (*N* = 89).

	Post Regulation Anxiety	Post Interview Anxiety	Post Mathematics Task Anxiety	Final Anxiety
	β	β	Β	β
Baseline anxiety	0.327 **	0.374 **	0.339 **	<0.001
Agreeableness	−0.013	−0.107	−0.149	−0.069
Extraversion	−0.022	−0.231 *	−0.254 *	0.010
Neuroticism	0.135	−0.153	−0.141	−0.027
ΔR^2^	0.021	0.063	0.082	0.005

* *p* < 0.05. ** *p* < 0.01.

## Data Availability

This study was not preregistered. All the data reported in the present manuscript are publicly available via Open Science Framework and can be accessed now at: https://osf.io/jsdve/?view_only=4eaee06c4cf0498b9f88d4d6cf7d21b3.

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
