# Peer review of "The Influence of Personality on Interpersonal Emotion Regulation in the Context of Psychosocial Stress"

_ijerph, 2023, doi:10.3390/ijerph20043073_

Round 1

Reviewer 1 Report

The paper describes a study on interpersonal emotion regulation and the relation between the personality of the regulator and the target's emotion regulation. The study evaluated the communication between pairs of participants who acted as the regulator and the target. It was observed that the extraversion had an influence on the amount of fluctuations in the target's anxiety levels. The paper is structured well with sufficient attention given to literature study, methods and analysis. One area where the paper can be improved is addition of the list of strategies used by the regulators. The paper does provide a few examples but a list of all the strategies used by various regulators would be useful to the readers (may be as part of supplementary material).  A few questions that can be clarified, are: 1) Whether the targets were aware of the interview topic? 2) What was the reason behind choosing a Math question as opposed to a puzzle or some other general knowledge question. Is there an existing study that can be cited, to support this step? The reviewer did have initial questions on the scenario and self reporting steps, however they have been satisfactorily identified as a limitation in the section 4.1. Overall the paper is well organized with sufficient details for repeatability and would make a good read on an interesting research topic. Therefore, the reviewer recommends an accept.

Author Response

Reviewer 1.

1.1. The paper is structured well with sufficient attention given to literature study, methods and analysis.

We thank the reviewer for their careful reading of our manuscript and their positive assessment of our work.

1.2. One area where the paper can be improved is addition of the list of strategies used by the regulators. The paper does provide a few examples but a list of all the strategies used by various regulators would be useful to the readers (may be as part of supplementary material). 

As discussed in our methods, we asked regulators to indicate the extent to which they used each of four different types of strategies. These were 1; attention modification, 2; Cognitive change, 3; Expressive suppression, and 4; socio-affective strategies. We were not able to observe and codify regulation strategies during the interaction. We have also noted in the limitations section that future studies would benefit from recording and coding strategy use.

1.3. Whether the targets were aware of the interview topic?

Targets were told that they would be engaging in a job interview for a fictious position at the University.

1.4. What was the reason behind choosing a Math question as opposed to a puzzle or some other general knowledge question. Is there an existing study that can be cited, to support this step?

The standardised Trier Social Stress task, which has been used in thousands of studies to elicit social stress, involves three components. The first is anticipatory stress (where participants prepare for an upcoming job interview), the second is the interview (conducted in front of an unresponsive panel), and the third is the mental arithmetic task. The latter task is stressful because it involves forced failure, which is known to increase levels of psychosocial stress (e.g., see meta-analysis on psychosocial stress by Dickerson and Kemeny, 2004). We followed this standardised procedure in our study.

Reviewer 2 Report

Issues taken up by the authors of interpersonal emotion regulation should be considered of particular interest. The article is well structured and the research procedure is well documented including the presentation of research results. However, the reviewer suggests making some minor adjustments.

1. In the summary at the beginning, it would be good to formulate the goal of the undertaken research project

2. The formulated conclusions seem very small and modest. From the thoroughly presented research results, a number of implications, both cognitive and application, can be formulated. The reviewer is of the opinion that it would be good to expand them and draw attention to the applicable nature of the data obtained. Reviewer suggests paying attention, for example, to the possibility of using it in a crisis intervention situation.

3. Reviewer points out including information about “Strengths, Limitations and Future Directions” at the end of the article after presenting the conclusions.

Author Response

Reviewer 2

2.1. In the summary at the beginning, it would be good to formulate the goal of the undertaken research project

We have now edited the introduction to read (pages 1-2):

“Therefore, the goal of this study was to assess whether particular aspects of the regulator’s personality influenced the process of interpersonal emotion regulation in the context of a psychosocial stressor. “

2.2. The formulated conclusions seem very small and modest. From the thoroughly presented research results, a number of implications, both cognitive and application, can be formulated. The reviewer is of the opinion that it would be good to expand them and draw attention to the applicable nature of the data obtained. Reviewer suggests paying attention, for example, to the possibility of using it in a crisis intervention situation.

We thank the reviewer for this comment. We have now added the following in their discussion on page 13:

“The finding that people who are higher in extraversion appear to be more able to deal with others’ anxiety suggests that such individuals may be well positioned in roles that involve this task, such as crisis intervention, debt management, counselling, and career coaching.”

2.3. Reviewer points out including information about “Strengths, Limitations and Future Directions” at the end of the article after presenting the conclusions.

The conclusions we formed took into account not only the findings of the study and our interpretation of them, but also our assessment of the strengths and limitations of the work, alongside the future efforts needed to build on our study. Therefore, we believe the order of the sections, with “strengths, limitations and future directions” coming before the conclusion of the paper, makes sense.

Reviewer 3 Report

The modern world is impossible to perceive without considering the stress factor. Although this phenomenon has been studied for quite a long time, and its concept, structure, definition and understanding in psychological science are widely represented, yet globalization, informatization and tension mounting in society dictate the necessity to research this construct. Both external social environment and individual-and-psychological features of the person impact his understanding of the stress and attitude towards it. Comprehending the specifics of functioning in a stressful situation of a particular individual, endowed with certain qualities and characteristics, can allow for shaping not only a scientific understanding of this process, but also for determining the direction of practical application of the results of this study, namely, the formation of adaptation programs for a particular type of person according to his/her subjective attitude to stress.

Questions and comments:

1. Was the situation of the interview game really stressful for students?

2. In order to prove the stress generated by the interview, it was necessary to conduct a study revealing the background stress level of students before the interview.

3. There are too many descriptions of why the results of the research are not sufficiently substantiated, which makes the results slip away and level out.

Author Response

Reviewer 3

3.1. Was the situation of the interview game really stressful for students?

The Trier Social Stress task is a very well established generator of stress and anxiety in participants. In the present study, we are confident that the interview situation was genuinely stressful because we saw a significant increase in reported anxiety levels between timepoints 1 and 2 (t = -6.6, p<0.001). Indeed, this is further supported by the high AUC scores, which summarize change in anxiety levels during the Trier task.  

3.2. In order to prove the stress generated by the interview, it was necessary to conduct a study revealing the background stress level of students before the interview.

We measured state levels of anxiety multiple times throughout the course of the study. This is most clearly demonstrated in Figure 1 from the manuscript, where each red circle on the figure represents an occasion when anxiety was captured. As you can see, our first measure precedes the first social stressor (the interview preparation period), meaning that we do capture background levels of anxiety.

3.3. There are too many descriptions of why the results of the research are not sufficiently substantiated, which makes the results slip away and level out.

We have tried to balance a cautious interpretation of our findings by not only highlighting the significant results, but considering why some of our hypotheses were not supported. We appreciate that this can feel like diluting the main message, but feel this is a necessary and honest way to conclude our findings.

Reviewer 4 Report

The manuscript under review investigated the relations between personality and the ability to effectively regulate others' emotions. The study paired "regulators" with "targets" and had the regulators try to manage the targets' feelings before a stressful job interview.  This is an original topic, and the study has the potential to contribute to the literature on intrapersonal emotion regulation.

Contrary to the hypotheses, the study found no relationship between the regulators' personality traits and their reported emotion regulation strategies, nor between the regulators' personality and the targets' job interview performance.  These negative findings are results themselves, assuming there is no bias in the experiment's methodology.

Therefore, the authors must reflect on and discuss why the expected results did not occur.

11)      One problem I see in conducting the experiment is that there is no “true” manipulation check. This is a major limitation because a possible reason for non-significant results could be that the manipulation was not equally effective for all the regulators.  From my reading of the manuscript, It is not clear whether the five minutes (in which the regulator should help the target person to be prepared for the interview) were recorded or transcribed verbatim, but if they were, it would be useful to have such data, for example, to see the quantity and quality of exchanges between the target and the regulator. It could be that in some cases the regulators interacted little with the target, or, in other cases, the interaction did not involve central aspects of intrapersonal emotion regulation.  The quantity and quality of interactions could be a moderator of the relationship between interpersonal emotion regulation and interview performance (e.g., extroverted might have been more effective because they are more active and did more things during the interaction with the target).

22)      Although the study uses a sufficiently large sample and uses a regression approach, the lack of a linear association between the regulator's personality and the target person's behavior could be nonlinear. We do not know where the regulators stand with respect to certain traits such as extraversion, agreeableness, or neuroticism. The effect of traits on behavior may require the person to possess a particularly high level of a specific trait to put trait-related elements of intra-personal emotional regulation in the interaction with the target person. For example, a regulator may have to be well above the average agreeableness level to be able to regulate the target person's anxiety. Maybe using a dummy categorization for each trait in regression analyses (e.g., medium vs. low and high vs low agreeableness) might be useful to test whether differences in the dependent variables arise only at very high levels of each trait, relative to a reference category).

33)      Restriction of range could be another reason for non-significant results. I see that the variability in agreeableness and neuroticism is rather low. There are two ways to address the restriction of range in correlation analysis. One way to address the restriction of range is using a non-parametric test, such as Spearman's rank-order correlation, which is less sensitive to the restriction of range than parametric tests such as Pearson's correlation. Another way to address the restriction of range is to transform the data using a function that expands the range of the data. For example, taking the logarithm of the data can sometimes expand the range and make the data more suitable for correlation analysis. I guess that restriction of range can be ruled out by comparing the present analyses with corresponding nonparametric alternatives or transforming the predictor set.

44)      From my reading of the article, it is not clear why the Interpersonal Emotion Regulation Management questionnaire was administered to the regulators even before the experimental session. It would be helpful to explain this detail and possibly use the data in the analyses.

55)      Similarly, it is not clear to me why the authors, prior to synthesizing the state anxiety data (with AUC), did not do descriptive analyses of the target person's state anxiety and did not consider using the individual measures in a separate serious of analyses (also as difference scores from baseline assessment). Relatedly, there might be target persons, who were trait anxious (obtaining high scores in all five measurements taken). For these individuals, it would be difficult to expect that five minutes of interindividual emotion regulation could be effective.

66)      The subjects of the experiment were sampled from the same university campus and probably attended the same introductory psychology courses. One threat to the internal validity of the research is that some regulator-target pairs already knew each other, or that some regulators knew the targets of a later session and vice versa (they may thus have exchanged impressions and opinions about the experiment) limiting the effectiveness of the experimental manipulation,

77)      The sex of the targets and regulators and whether they were same-sex or different-sex dyads should be considered as moderators in the regression analyses.

I hope the authors will consider the above points as pointers for improving their manuscript, which in any case in my opinion appears to have sufficient scientific quality.

Author Response

4.1. One problem I see in conducting the experiment is that there is no “true” manipulation check. This is a major limitation because a possible reason for non-significant results could be that the manipulation was not equally effective for all the regulators.  From my reading of the manuscript, It is not clear whether the five minutes (in which the regulator should help the target person to be prepared for the interview) were recorded or transcribed verbatim, but if they were, it would be useful to have such data, for example, to see the quantity and quality of exchanges between the target and the regulator. It could be that in some cases the regulators interacted little with the target, or, in other cases, the interaction did not involve central aspects of intrapersonal emotion regulation.  The quantity and quality of interactions could be a moderator of the relationship between interpersonal emotion regulation and interview performance (e.g., extroverted might have been more effective because they are more active and did more things during the interaction with the target).

We thank the reviewer for this point. We did not record the interaction, and so are unable to objectively quantify the quality of interactions. We had already noted this point as a suggestion for future research in our discussion section, as we agree it would be really helpful. We have now added to this text to explicitly recognise your point, which can be seen on page 14:

“A further advantage of video-recording interactions is that we would gain insight into additional personality-related differences in interpersonal emotion regulation aside from strategy use that could explain why regulators high in extraversion are apparently more effective at regulating others’ anxiety (e.g., do they interact more with the target, or express more positive affect?)”

4.2. Although the study uses a sufficiently large sample and uses a regression approach, the lack of a linear association between the regulator's personality and the target person's behavior could be nonlinear. We do not know where the regulators stand with respect to certain traits such as extraversion, agreeableness, or neuroticism. The effect of traits on behavior may require the person to possess a particularly high level of a specific trait to put trait-related elements of intra-personal emotional regulation in the interaction with the target person. For example, a regulator may have to be well above the average agreeableness level to be able to regulate the target person's anxiety. Maybe using a dummy categorization for each trait in regression analyses (e.g., medium vs. low and high vs low agreeableness) might be useful to test whether differences in the dependent variables arise only at very high levels of each trait, relative to a reference category).

We agree with the reviewer that this work seeks to identify linear relationships between personality factors and interpersonal emotion regulation approach. We do not feel that taking an arbitrary cut-off for these continuous variables would be beneficial given our apriori hypotheses. This could of course be examined in future studies, particularly with larger sample sizes and more exploratory approaches.

4.3. Restriction of range could be another reason for non-significant results. I see that the variability in agreeableness and neuroticism is rather low. There are two ways to address the restriction of range in correlation analysis. One way to address the restriction of range is using a non-parametric test, such as Spearman's rank-order correlation, which is less sensitive to the restriction of range than parametric tests such as Pearson's correlation. Another way to address the restriction of range is to transform the data using a function that expands the range of the data. For example, taking the logarithm of the data can sometimes expand the range and make the data more suitable for correlation analysis. I guess that restriction of range can be ruled out by comparing the present analyses with corresponding nonparametric alternatives or transforming the predictor set. Non-parametric analyses would be particularly useful where there is a non-normal distribution of data. Our personality data was all normally distributed and so we retain more power by using parametric correlations (given that all of the assumptions were met).

There is nothing to suggest that we recruited individuals with a restricted range of neuroticism or agreeableness. We used open recruitment. After a review of other studies who sample British undergraduate psychology students we can confirm that the means and SDs of students’ agreeableness and neuroticism are comparable in that trait agreeableness tends to be high on average with lower variability for agreeableness and neuroticism (e.g., see Chamorro-Premuzic & Furnham, 2009). Therefore, there is no evidence to suggest that our sample suffered from range restriction as values were in line with those reported in other studies sampling UK undergraduate psychology students.

4.4.From my reading of the article, it is not clear why the Interpersonal Emotion Regulation Management questionnaire was administered to the regulators even before the experimental session. It would be helpful to explain this detail and possibly use the data in the analyses.

We apologise for the confusion. The interpersonal emotion management questionnaire was administered after the regulation period. We have now edited Figure 1 (attached as a word document) to clarify this.

4.5. Similarly, it is not clear to me why the authors, prior to synthesizing the state anxiety data (with AUC), did not do descriptive analyses of the target person's state anxiety and did not consider using the individual measures in a separate serious of analyses (also as difference scores from baseline assessment). Relatedly, there might be target persons, who were trait anxious (obtaining high scores in all five measurements taken). For these individuals, it would be difficult to expect that five minutes of interindividual emotion regulation could be effective.

We agree that collecting multiple measures of anxiety as we did affords a diverse range of analysis opportunities. For this reason, and following convention in the literature (e.g., Pruessner et al., 2003), we committed to a strategy for testing our hypotheses a priori that adopted the area-under-curve index of anxiety measures. This index was preferred because it synthesises the multiple measures and therefore maximises the information we can use within a single test of the hypotheses.

Nevertheless, you are quite right that the data could be analysed in other ways. To explore what would happen if we used the separate anxiety measures rather than the area-under-curve index, we ran correlations using the separate anxiety measures, and then a series of regression analyses predicting the separate anxiety measures in which we controlled for baseline anxiety and included all three personality traits as predictors. Across both sets of analyses, there were no relationships between the personality traits and the baseline anxiety measure. For the second anxiety measure, taken immediately after the period of regulation, the regulator’s neuroticism was negatively related to this, but only in the raw correlations (i.e., after controlling for the target’s baseline anxiety and in the presence of the other personality traits, this relationship disappeared). For the third anxiety measure, taken immediately after the social stressor of the job interview, there was a negative correlation with extraversion (r = -.26, p < .05) that remained significant in the regression analysis (β = -.23, p < .05). For the fourth anxiety measure, taken just after the mathematics task, we observed the same, i.e., there was a negative correlation with extraversion (r = -.21, p < .05) that remained significant in the regression analysis (β = -.25, p < .05). There were no relationships between personality traits and the final anxiety measure at the end of the study period.

Because these findings support and slightly extend our understanding based on the original analyses, we have added the following to the paper, presenting these as exploratory rather than hypothesis-testing analyses, on pages 11-13:

“Additional exploratory analyses attempted to pinpoint the specific times at which regulators’ extraversion was making a difference to targets’ anxiety. We conducted a series of regressions with the raw anxiety measures (from the different time points across the study period) as dependent variables, baseline anxiety included as a control variable in Step 1, and the three personality traits as predictors in Step 2. The results, in Table 4, show that for targets who were paired with regulators who were higher in extraversion, they experienced lower levels of anxiety (controlling for baseline) after both stressors (i.e., the interview and mathematics task). These results confirm that the anxiety of targets who were paired with more extraverted regulators was better managed during the psychosocial stressor they encountered and that those targets experienced further protection against the subsequent mathematical stressor.” 

In terms of individual differences in target anxiety, this is a good point. There isn’t a cut-off for ‘high anxiety’ that we are aware of, to code/identify these people in our data, but we now raise this as an issue for future research on page 14:

“Further individual differences, such as those relating to the target of regulation (e.g., their general anxiety levels), could also be explored.”

4.6. The subjects of the experiment were sampled from the same university campus and probably attended the same introductory psychology courses. One threat to the internal validity of the research is that some regulator-target pairs already knew each other, or that some regulators knew the targets of a later session and vice versa (they may thus have exchanged impressions and opinions about the experiment) limiting the effectiveness of the experimental manipulation,

While this is a possibility, our recruitment pool covers more than 900 students from across two psychology cohorts and other university courses, so we believe it is unlikely that any of the randomly selected pairs would be well acquainted. We also do not think it would have threatened the validity of the study if a participant in one study session were to tell a participant in a later session about the study, albeit that this is an unlikely event to have happened, because there was no deception involved in the research.  

4.7. The sex of the targets and regulators and whether they were same-sex or different-sex dyads should be considered as moderators in the regression analyses.

As can be seen from the intercorrelations between study variables in Table 1, neither the sex of the regulator or target was systematically related to any of the study variables. However, we did not previously consider if the status of the dyad as same versus different sex would relate to any study variables. After checking our data, we can confirm that of the 89 dyads, 61 were same-sex. We created a new variable to represent whether the dyad was same versus different sex and found that this variable was unrelated to all of our outcomes (regulation strategy use, area-under-curve anxiety, target interview performance), rs .01-.18. Given the above, we do not believe that sex is a relevant factor in our study.

Round 2

Reviewer 4 Report

Thank you for taking the time to revise your manuscript based on my previous comments. I have taken the time to read the revised manuscript. I commend the authors for their willingness to accept constructive criticism.

Two minor comments:

"We agree with the reviewer that this work seeks to identify linear relationships between personality factors and interpersonal emotion regulation approach. We do not feel that taking an arbitrary cut-off for these continuous variables would be beneficial given our apriori hypotheses. This could of course be examined in future studies, particularly with larger sample sizes and more exploratory approaches."

Acknowledge this in the discussion.

 "There is nothing to suggest that we recruited individuals with a restricted range of neuroticism or agreeableness. We used open recruitment. After a review of other studies who sample British undergraduate psychology students we can confirm that the means and SDs of students’ agreeableness and neuroticism are comparable in that trait agreeableness tends to be high on average with lower variability for agreeableness and neuroticism (e.g., see Chamorro-Premuzic & Furnham, 2009). Therefore, there is no evidence to suggest that our sample suffered from range restriction as values were in line with those reported in other studies sampling UK undergraduate psychology students."

This is important information. State this in the discussion.

Author Response

1.1. "We agree with the reviewer that this work seeks to identify linear relationships between personality factors and interpersonal emotion regulation approach. We do not feel that taking an arbitrary cut-off for these continuous variables would be beneficial given our apriori hypotheses. This could of course be examined in future studies, particularly with larger sample sizes and more exploratory approaches." Acknowledge this in the discussion.

1.1.This has now been recognised in the discussion on page: 12-13:

“A final suggestion for future research is to explore the possibility of non-linear relationships between personality and interpersonal emotion regulation.“

1.2. "There is nothing to suggest that we recruited individuals with a restricted range of neuroticism or agreeableness. We used open recruitment. After a review of other studies who sample British undergraduate psychology students we can confirm that the means and SDs of students’ agreeableness and neuroticism are comparable in that trait agreeableness tends to be high on average with lower variability for agreeableness and neuroticism (e.g., see Chamorro-Premuzic & Furnham, 2009). Therefore, there is no evidence to suggest that our sample suffered from range restriction as values were in line with those reported in other studies sampling UK undergraduate psychology students." This is important information. State this in the discussion.

1.2 This has now been recognised in the discussion on page: 13-14:

"A final possibility is that the pattern of null results in this regard could be an issue of range restriction stemming from the low observed variability in the traits of agreeableness and neuroticism in our sample. However, the means and standard deviations for personality traits in out sample are comparable to other studies sampling British undergraduate psychology students (e.g., [56]), meaning that our sample is likely to be representative of the wider population from which it is drawn."